# Virtual Reality Simulation-Based Clinical Procedure Skills Training for Nursing College Students: A Quasi-Experimental Study

**DOI:** 10.3390/healthcare12111109

**Published:** 2024-05-29

**Authors:** Hyeongyeong Yoon, Eunwha Lee, Chung-Jong Kim, Yoonhee Shin

**Affiliations:** 1Seongnam Campus, College of Nursing, Eulji University, Seongnam-si 13135, Republic of Korea; hkyoon@eulgi.ac.kr; 2Department of Nursing, Ewha Womans University Seoul Hospital, Seoul 07804, Republic of Korea; 40031s@eumc.ac.kr; 3Department of Internal Medicine, Ewha Womans University College of Medicine, Seoul 07985, Republic of Korea; erinus@ewha.ac.kr; 4Ewha Medical Academy, Ewha Womans University Seoul Hospital, Seoul 07985, Republic of Korea; 5College of Nursing, Ewha Womans University, Seoul 03760, Republic of Korea

**Keywords:** virtual reality, nursing students, clinical competence, satisfaction, confidence

## Abstract

Core nursing skills are emphasized in nursing education, given their vital role in nurses’ competence; however, invasive nursing procedures like catheterization and enemas are infrequently performed in actual clinical practice, primarily being observed rather than executed. Virtual reality simulation training involves performing core nursing skills on virtual patients in a three-dimensional virtual reality environment, following the correct procedures. The purpose of this study is to examine the effects of VR simulation on nursing students’ confidence, proficiency, task engagement, and satisfaction. The study participants included 76 second-year nursing students, with 37 in the VR group and 39 in the control group. The VR group engaged in immersive VR (IVR) training sessions including enemas, nasogastric feeding, and nelaton catheterization. Conversely, the control group practiced these skills using mannequins. Assessments evaluated confidence, proficiency, learning satisfaction, and task engagement before and after the intervention. The average age of the study participants was 21.07 years, with 78.95% being female and 21.05% being male. The study findings revealed no significant differences between the VR and control groups regarding confidence (F = 3.878, *p* = 0.053), task engagement (F = 0.164, *p* = 0.687), and learning satisfaction (F = 0.668, *p* = 0.416). However, the VR group demonstrated significantly higher proficiency in the overall assessment of nasogastric feeding (F = 5.389, *p* = 0.023) and core components of nelaton catheterization (F = 4.046, *p* = 0.048). The IVR program emerged as an effective and valuable teaching tool, particularly well-suited for second-year nursing students, significantly enhancing proficiency.

## 1. Introduction

Virtual reality (VR) simulation is a technology to generate a 3D virtual world, providing users with an immersive experience akin to being in the real world [1]. VR can be categorized into three levels of immersion: non-immersive, semi-immersive, and fully immersive VR based on the concept of extended reality [2]. Non-immersive VR, also known as desktop VR, involves the creation of a 2D virtual environment displayed on a computer screen. In contrast, immersive VR immerses users in a virtual world, engaging senses such as vision and hearing within a 3D virtual space [2,3]. Recent advancements in VR technology have shifted the focus from 2D to 3D experiences, with hardware improvements leading to the release of high-quality products featuring high display resolutions, thus enhancing the realism of the immersive experience [4].

Core nursing skills are essential as they are frequently used in the daily responsibilities of nurses and constitute a fundamental component of nursing education and training [5,6]. Core nursing skills are frequently used in the daily tasks of nurses and are fundamental elements of nursing education and training [5,6]. It has been shown that nursing students’ experience with essential nursing skills enhances their clinical performance and leads to higher levels in education and collaboration, interpersonal relationships and communication, and professional development areas [7]. The Korean Accreditation Board of Nursing Education (KABONE) identifies 18 core nursing skills, including enemas, catheterization, and intravenous infusion, as integral to nursing education. It mandates nursing schools to incorporate these skills into their practical education programs and evaluate and oversee student performance until graduation [6]. Despite the heightened emphasis on core nursing skills within the nursing curriculum, invasive nursing procedures like catheterization and enema are infrequently performed in actual clinical practice, primarily being observed rather than executed [7,8]. The experience and frequency of practicing core nursing skills hold significant importance in determining clinical performance [7,9]. However, new nurses often exhibit low confidence levels when it comes to core nursing skills, except for vital signs [10]. To address this issue, educational methods are required to enhance students’ hands-on experience in performing core nursing skills within an environment closely resembling real clinical settings. Nursing education incorporating virtual simulation emerges as an effective approach, enabling students to safely acquire these skills within a clinical-like 3D environment [11,12,13]. VR offers distinct advantages in this context, enabling repetitive skill practice without constant instructor supervision while providing an engaging and enjoyable learning experience [11,14]. Research indicates that 96.2% of new nurses found simulation training for core nursing skills instrumentally helpful in boosting their confidence and proficiency in nursing [15]. Moreover, there exists a direct relationship between the frequency of performing core nursing skills among nursing students and their level of confidence [9,16]. Given that VR facilitates repetitive practice, it stands as an educational method capable of enhancing students’ confidence. Furthermore, it has been demonstrated that higher confidence and increased learning engagement translate into improved clinical performance [3,17,18].

To date, VR nursing education has employed a combination of desktop VR and IVR, with a predominant focus on 2D VR education using desktop systems [3,19]. For example, in the context of nasogastric feeding VR simulations, the emphasis has been on knowledge acquisition through quizzes, anatomical images, and scenarios using VR, rather than comprehensive technical training [19]. Similarly, intravenous infusion education using VR has primarily involved immersive anatomy viewers and haptic devices through desktop systems [20]. In the case of nelaton catheterization VR training, the focus has been on repetitive catheterization practice within a computer game format, which does not cover the complete range of technical procedures from preparation to treatment [21]. Moreover, when considering 3D VR studies for nursing students, there have been investigations into specific procedures such as endotracheal intubation [22] and transfusion therapy [23]. However, it is worth noting that these studies typically evaluated the effects after only a single session of 3D VR training. In both mentioned studies, the intervention periods were relatively short, making it challenging to draw broad conclusions regarding the effectiveness of VR training solely based on post-test scores. To address this limitation, our study extended the IVR core nursing skills program over six weeks, allowing ample time for participants to practice these skills within a virtual ward environment closely resembling real clinical settings. This study focused on enemas and nasogastric feeding as a priority for VR program development, based on identified needs [5]. We aimed to evaluate the confidence, proficiency, task engagement, and learning satisfaction of 3D VR education by implementing three IVR programs covering enemas, nasogastric feeding, and nelaton catheterization. 

## 2. Materials and Methods

### 2.1. Study Design

This study employed a non-equivalent control group pre–post experimental design to assess the impact of an IVR program on nursing students’ performance confidence, proficiency, task engagement, and learning satisfaction while focusing on core nursing skills such as enemas, nasogastric feeding, and nelaton catheterization.

### 2.2. Participants

The study participants comprised sophomore students enrolled in the nursing department of A University in South Korea. The study duration spanned six weeks, from 3 May 2023, to 7 June 2023. Sample size was determined using G*Power 3.1.9.7, with the effect size derived from a previous study that applied a VR simulation program to nursing students, confirming its effectiveness and usability [24]. The established effect size was 0.83. Considering a two-sided test, a significance level of 0.05, 90% power, and an equal number of participants in both groups (n1 = n2), the minimum required sample size was determined to be 32 students per group. In this study, two out of five fundamental nursing practice classes were assigned to the experimental group, while two other classes were designated as the control group. Before the study’s commencement, only students who voluntarily opted to participate were included. Two students were excluded from the initial pool of 80 students, resulting in a final cohort of 39 students in both the experimental and control groups. Two students in the experimental group dropped out, leaving 37 students in the experimental group and 39 in the control group to complete and provide data for analysis (Figure 1). Inclusion criteria necessitated that the participants be sophomore students concurrently enrolled in Fundamental Nursing I, having never received prior instruction in enema, nasogastric feeding, or nelaton catheterization. Exclusion criteria included individuals with conditions such as dizziness, Meniere’s disease, and headaches.

### 2.3. Intervention

Both groups comprised students enrolled in a nursing school that offers practical training in core nursing skills using mannequins. All participants underwent identical theoretical and practical classes covering enema, nelaton catheterization, and nasogastric feeding as part of their fundamental nursing practice curriculum. The control group did not receive any additional training in core nursing skills or VR instruction beyond their standard hands-on training, which involved using mannequins. After a six-week interval, the control group completed a post-survey with the same content as the pre-survey. In contrast, the experimental group was introduced to an IVR program for these three core nursing skills. This VR program was implemented at A General Hospital in Seoul, Korea, where the participants received comprehensive instruction, including an overview of the program and guidance on utilizing the VR devices. Following this initial training session, the experimental group engaged in VR training sessions once a week for five consecutive weeks, starting from the date of their initial training. In the sixth week following the application of the IVR program, the experimental group completed a post-survey, identical to the pre-survey.

### 2.4. Control Group

The control group began by completing demographic surveys, along with confidence, proficiency, task engagement, and learning satisfaction, before their practical training sessions. During these training sessions, students worked in pairs. One student practiced on a mannequin, while the other assessed skill levels and measured the time taken for each procedure using a proficiency checklist. The control group practiced these core nursing skills once a week for five weeks, with each skill practice session lasting 10 min, totaling 30 min per skill. One week after concluding the five-week practice, a follow-up survey was administered to the control group to assess any changes in their confidence, proficiency, engagement, and satisfaction levels resulting from the training sessions.

### 2.5. Experimental Group

The experimental group began by undergoing pre-intervention orientation and training on how to use the VR devices. Like the control group, they also completed surveys covering demographic, confidence, proficiency, task engagement, and learning satisfaction. Participants in the experimental group visited the VR simulation room at A General Hospital in Seoul once a week for a total of five weeks. During these sessions, they engaged in IVR training for enemas, catheterization, and nasogastric feeding. The VR skill training was led by the responsible professor, while device usage was overseen by two researchers and lasted for approximately 30 min. For those students encountering difficulties with operating the device, additional one-on-one training sessions were conducted until they became proficient. Once participants fully grasped device operation, they practiced the nursing skills independently while wearing a VR headset (Oculus Quest) and utilizing two controllers. The VR program employed in this study was the nursing education VR content, a collaborative effort between the VR company VRAD (VRAD Corporation, Hanam-si, Korea) and Ewha Womans University Medical Center. To create an immersive practice environment, the VR program presents the learner with a 3D VR background closely resembling a real hospital setting through a VR headset. The medical supplies used in the program are modeled on the actual supplies utilized in performing core nursing skills in a hospital setting. When the learner selects a specific core nursing skill for practice in VR, the VR headset initially displays detailed information. This includes the skill’s achievement goals, prerequisite knowledge, the required equipment and supplies, and the target time and difficulty level associated with the core nursing skill. The learner then proceeds to perform the selected skill, operating the controller while following step-by-step instructions aligned with the procedure. Immediate feedback is provided based on the learner’s performance, with the requirement that each procedure must be completed flawlessly to progress to the next level. The duration of the VR programs is set at 10 min for each skill. If the VR practice session extends beyond 30 min, the program pauses, and a 10-min break is included to mitigate the potential for cybersickness. At Week 6, one week after the conclusion of the five-week program, a survey on confidence, proficiency, engagement, and satisfaction was administered. 

### 2.6. Research Tools

#### 2.6.1. Confidence

To subjectively assess the level of confidence in nursing skills, the practitioner confidence tool developed by Grundy [25] was utilized. This tool comprises five items, each rated on a 5-point Likert scale, with scores ranging from 5 to 25. Higher scores indicate a greater level of confidence in nursing practice. In the context of this study, Cronbach’s alpha coefficient for this tool was calculated to be 0.940, demonstrating its reliability.

#### 2.6.2. Proficiency

Proficiency in the three core skills was evaluated as outlined in the Korean Nursing Education Evaluation Center Core Fundamental Nursing Skills Evaluation Items (Version 4.1). In the checklist for each evaluation, essential skills are marked as *important. The items were composed including the items marked with an *. The proficiency assessment instrument comprised 10 questions, including each core performance item specified for core fundamental nursing skills. Each item was evaluated using a 5-point Likert scale. Higher scores on this scale correspond to higher proficiency levels. The Cronbach’s alpha coefficients for both the total proficiency assessment tool and the core items of each skill assessment tool were calculated as follows: Enema: Total = 0.926, Core = 0.856; Nelaton Catheterization: Total = 0.932, Core = 0.892; and Nasogastric Feeding: Total = 0.966, Core = 0.953.

#### 2.6.3. Task Engagement

Task engagement state was assessed using the Korean version of the Flow State Scale for Occupational Task (K_FSSOT). This scale was originally developed by Yoshida et al. [26] and subsequently revised by Lee and Park [27] to ensure its validity and reliability. It comprises 14 questions across three domains and is evaluated using a 7-point Likert scale. The total score on this scale ranges from 14 to 98 points. In this study, the reliability was 0.927.

#### 2.6.4. Learning Satisfaction

To assess educational learning satisfaction with the provided VR nursing education program, a modified learning satisfaction evaluation tool was employed [28]. It is measured using a 5-point Likert scale and comprises five statements, in addition to three open-ended questions. For the study’s purpose, learning satisfaction was assessed using the five questions from the tool, excluding the open-ended questions. The total score on this modified tool ranges from 5 to 25 points, with higher scores indicating a higher level of learning satisfaction. The Cronbach’s alpha coefficient was calculated to be 0.936.

### 2.7. Analysis

The data were analyzed using SPSS PC+ 22.0 for Windows (Armonk, NY, USA). To describe the general characteristics of the study participants, frequencies and percentages were employed. For the evaluation of pre- and post-test differences in nursing confidence, proficiency, task engagement, and learning satisfaction within each group, paired *t*-tests were conducted. To assess differences between the experimental and control groups in terms of nursing confidence, proficiency, task engagement, and learning satisfaction, an analysis of covariance (ANCOVA) was employed.

### 2.8. Ethical Considerations

This study underwent a comprehensive review and received approval from the Institutional Review Board of A University Hospital, under the reference number SEUMC203-04-004-003.

## 3. Results

### 3.1. Participant Homogeneity Test

The homogeneity test conducted on the general characteristics of the participants in both the experimental and control groups revealed a statistically significant difference (*p* < 0.001) in the gender distribution, with a higher number of male students in the experimental group (Table 1). However, aside from this gender difference, the two groups were otherwise homogeneous in terms of their general characteristics. Among the preliminary assessments of nursing confidence, proficiency, task engagement, and learning satisfaction for both the experimental and control groups, differences were observed in nursing confidence (*p* = 0.002) and task engagement (*p* = 0.039) (Table 2).

### 3.2. Confidence

Initially, both the experimental and control groups exhibited higher confidence levels in nursing performance during the pre-test. Consequently, when the post-test scores were analyzed with the pre-test scores covaried, the difference between the control group and the experimental group was not found to be statistically significant (F = 3.878, *p* = 0.053) (Table 3).

### 3.3. Proficiency

The experimental group achieved a statistically significant higher score in the overall item of nasogastric feeding (F = 6.581, *p* = 0.012) (Table 3) (Figure 2). Additionally, when focusing on the core items specified by the Nursing Education Evaluation Center for each core skill, statistically significant differences between the two groups were observed in nelaton catheterization (F = 4.046, *p* = 0.048) and nasogastric feeding (F = 5.389, *p* = 0.023) (Table 3) (Figure 3).

### 3.4. Task Engagement

In the pre-test, the experimental group demonstrated higher levels of task engagement compared to the control group. However, when the pre-test scores were considered in the analysis of post-test results, the difference between the control group and the experimental group was not found to be statistically significant (F = 0.164, *p* = 0.687) (Table 3).

### 3.5. Learning Satisfaction

There were no statistically significant differences in learning satisfaction between the experimental and control groups (F = 0.668, *p* = 0.416) (Table 3).

## 4. Discussion

With recent technological advancements, 3D IVR education has emerged as a promising complement to conventional nursing education methods. It is increasingly considered an approach to clinical practice education, particularly given the reduced traditional clinical practice opportunities [13]. 

The study findings highlight a statistically significant increase in the proficiency of the experimental group, particularly in the areas of nelaton catheterization and nasogastric feeding. A qualitative study on nursing graduates who experienced VR supported this study by indicating that VR enhances nursing knowledge and skills [29]. A relevant study that employed aspiration VR with second-year nursing students lends support to these results. In that study, the average score of the experimental group was 11.37, while the control group averaged 10.70 [22]. However, it is important to note that the referenced study utilized a non-equivalent control group post hoc design, solely comparing post-intervention scores following a single 30 min VR simulation training session. Thus, it does not provide insights into potential disparities in baseline skill performance. Consequently, there are limitations in generalizing these findings. In another study involving VR simulation for indwelling catheterization among nursing students, both the experimental and control groups demonstrated improvements in their skill scores. However, there was no discernible difference between the two groups [12]. Park’s study had a limitation in which students who expressed a preference for VR were placed in the experimental group, while others were assigned to the control group, resulting in no control for individual VR preferences. The significance of the current study lies in the controlled assignment of experimental and control groups based on class, effectively managing VR preference as a variable. Over the 6-week intervention period, proficiency scores for nelaton catheterization and nasogastric feeding improved from 82.7 to 85.2 points in the experimental group and from 81.9 to 84.6 points in the control group, demonstrating a statistically significant proficiency enhancement. It is worth noting that the control group’s post-test scores were lower than their pre-test scores, potentially attributed to their initial lower scores compared to the experimental group. Previous research has indicated that practicing core nursing skills using VR leads to more extended skill retention compared to practicing without instructor feedback in an open lab [21]. Therefore, future research should consider comparing the educational effectiveness of VR training with traditional instructor-led hands-on training. However, the improvement in enema proficiency scores was observed in both the experimental and control groups, with no statistically significant difference between them. This could be because enemas are part of the practical examination in the fundamental nursing practice curriculum, scheduled for one week after the intervention’s conclusion. Consequently, both groups may have increased their practice frequency, leading to improved proficiency scores. Future studies are advised to control for skills related to planned examination items to better understand the impact of VR training.

In this study, there were no statistically significant differences in terms of satisfaction, confidence, and engagement between the experimental and control groups. The engagement and satisfaction of senior nursing students who participated in the online virtual reality program were statistically significantly higher in the experimental group compared to the control group [30]. This study applied a virtual reality program over a total of 5 weeks and conducted a follow-up survey one week after the intervention. However, studies such as Yang et al. [30] conducted surveys immediately after a single VR simulation session, which only measured short-term effects. This presents limitations such as difficulty in generalizing the follow-up survey results due to external validity issues. Notably, this study was conducted within the framework of the independent practice time allotted for the fundamental nursing practice course, with five classes following the same curriculum. However, one key variation was the division of practical classes among three different instructors. This instructor variability could have influenced student satisfaction and confidence levels, as previous research has suggested that these factors are sensitive to instructor skills and teaching methodologies [31,32]. Previous studies on VR have indicated that IVR environments can effectively immerse nursing students in realistic hospital scenarios [23]. Nevertheless, there are technical challenges, such as students’ inexperience with VR controllers [3,23], that can potentially hinder the full immersion experience. Based on the results of a qualitative study [33] (Kim Yoon-Jung, Kim Won-Jeong, Min Hye-Young, 2020), some students felt confusion in the virtual environment and frustration from the scores received at the end of the virtual simulation. It is presumed that students who received low scores due to mistakes from unfamiliar movements experienced lower confidence, engagement, and satisfaction after the VR essential skills evaluation. Previous studies conducted on senior nursing students showed high levels of satisfaction and engagement [30]. However, this study targeted sophomore students who were learning basic nursing skills for the first time, indicating that the difference in academic year could have also influenced the results. In this study, VR operation methods were introduced during the orientation session; however, it is plausible that students encountered difficulties in executing detailed operations while practicing various core skill procedures. When introducing virtual reality simulations into the educational curriculum, students may encounter confusion with unfamiliar computer interfaces and lack of proficiency in handling them [33]. This can create a perception of a high barrier to virtual reality due to the reduced internal validity in proving the educational effectiveness of virtual reality education. Therefore, it is recommended to extend the time for students to become familiar with VR devices and programs for further research. Additionally, considering external factors such as exam periods and test items during regular course sessions may influence the outcomes, so it is advised to conduct research with regard to exam periods and test items.

### Limitations

This study was conducted with nursing students at a single university, which limits the generalizability of the results. The fundamental nursing practice education was distributed across five classes, with these classes being divided among three different instructors, introducing a limitation related to instructor control. Additionally, the experimental and control groups were not randomly assigned, and students could choose their preferred teaching method in two classes, resulting in disparities in the preliminary homogeneity of these two groups. 

## 5. Conclusions

In contemporary clinical practice, nursing students often have limited opportunities to actively practice these skills and are more frequently relegated to observational roles. Therefore, innovative teaching methods are essential to bridge this gap. The 3D immersive VR core nursing skills training program allows students to practice these skills following standardized procedures within a virtual ward environment, mirroring a real hospital setting. The results of the study showed that proficiency in nelaton catheterization and nasogastric feeding was statistically significantly improved. Hence, 3D VR training is strongly recommended as a highly effective and practical approach to enhancing nursing skills. Moreover, its self-paced nature allows students to engage in independent practice without the need for additional personnel, making it a valuable educational method for the technologically proficient generation.

## Figures and Tables

**Figure 1 healthcare-12-01109-f001:**
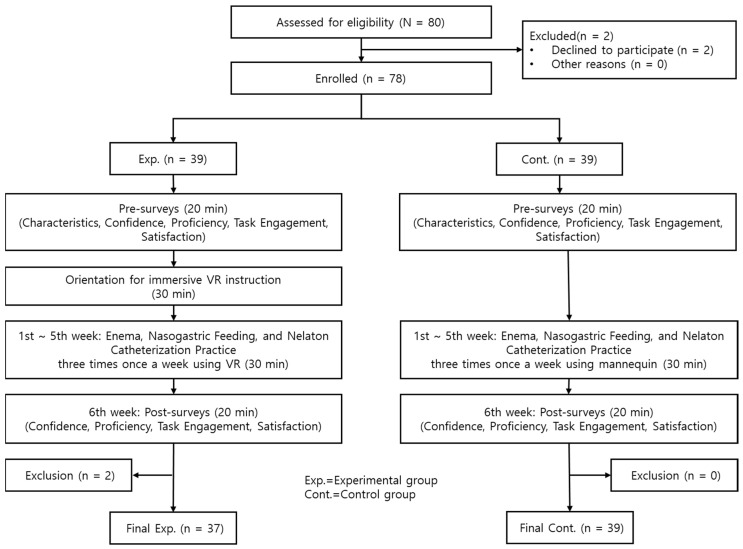
Research process.

**Figure 2 healthcare-12-01109-f002:**
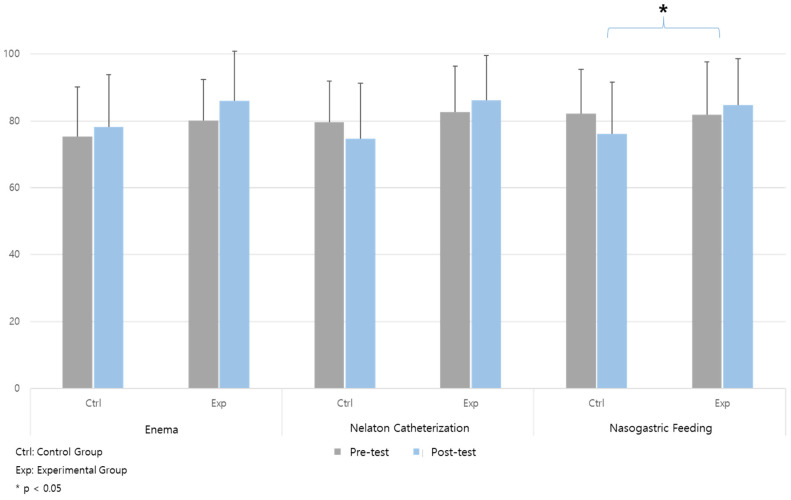
Proficiency: all items.

**Figure 3 healthcare-12-01109-f003:**
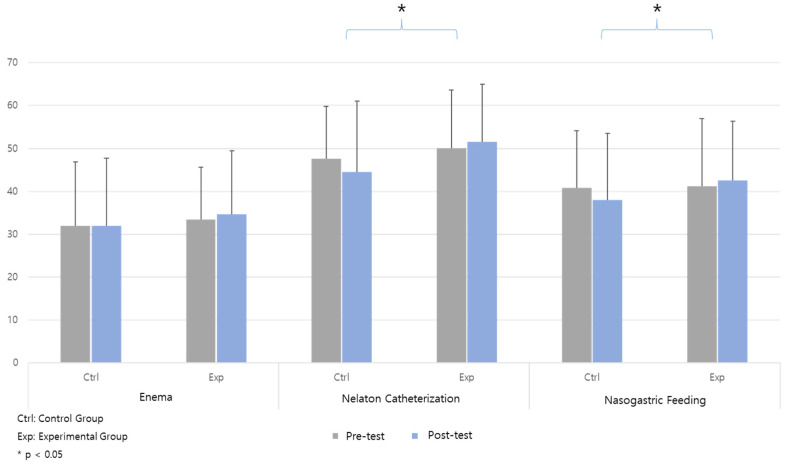
Proficiency: core items.

**Table 1 healthcare-12-01109-t001:** General characteristics.

Variable	Category	Total (*n* = 76)		Control (*n* = 39)		Experimental (*n* = 37)		*p*
		*N* or Mean	% or SD	*N* or Mean	% or SD	*N* or Mean	% or SD	
Gender	Male	16	21.05	0	0	16	43.24	<0.001 *
	Female	60	78.95	39	100.00	21	56.76	
Age (year)		21.07	±2.87	21.54	±3.79	20.57	±1.21	0.135
Learning Preference	Lecture	39	51.32	20	51.28	19	51.35	1.000
	Discussion	6	7.89	3	7.69	3	8.11	1.000
	Q&A	6	7.89	3	7.69	3	8.11	1.000
* *Multiple response*	Practical training	40	52.63	20	51.28	20	54.05	0.823
Last Semester GPA	<3.0	15	20.27	7	18.92	8	21.62	0.890
	3.0~4.0	48	64.86	25	67.57	23	62.16	
	≥4.0	11	14.86	5	13.51	6	16.22	
Nursing Skills-Related Educational Experience	Yes	67	88.16	35	89.74	32	86.49	0.733
	No	9	11.84	4	10.26	5	13.51	
Nursing Education-Related VR Experience	Yes	8	10.53	5	12.82	3	8.11	0.712
	No	68	89.47	34	87.18	34	91.89	
VR Experience Outside of Nursing Education	Yes	8	10.67	2	5.26	6	16.22	0.153
	No	67	89.33	36	94.74	31	83.78	
Purpose of VR Experience Other Than Nursing Education	Game	5	55.56	2	66.67	3	50.00	0.286
	Education/Learning	1	11.11	1	33.33	0	0	
	Cultural experience	3	33.33	0	0	3	50.00	
Average Time Spent on Smart Devices per Day (hour)		7.05	±2.71	7.38	±2.42	6.69	±2.98	0.266
Smart Device Utilization Skills	Not at all	0	0	0	0	0	0	0.444
	Not good	1	1.32	1	2.56	0	0	
	Moderate	13	17.11	7	17.95	6	16.22	
	Good	37	48.68	21	53.85	16	43.24	
	Very good	25	32.89	10	25.64	15	40.54	
Most Used Smart Devices	Smart phone	56	73.68	31	79.49	25	67.57	0.301
	Smart tablet	20	26.32	9	23.08	11	29.73	0.606
	Laptop	3	3.95	2	5.13	1	2.70	1.000
	Smart watch	0	0	0	0	0	0	1.000
* *Multiple response*	Smart TV	1	1.32	0	0	1	2.70	0.487
VR Device Ownership	Yes	3	4.00	3	7.89	0	0	0.240
	No	72	96.00	35	92.11	37	100.00	

Note: GPA= grade point average; VR = virtual reality. * *p* < 0.05.

**Table 2 healthcare-12-01109-t002:** Pre-test homogeneity test (dependent variable).

Variables			Total (*n* = 76)	Control Group (*n* = 39)	Experimental Group (*n* = 37)	t (*p*)
			Mean, SD	Mean, SD	Mean, SD	
Nursing Performance Confidence			15.61 ± 4.20	14.18 ± 4.10	17.11 ± 3.81	0.002 *
Proficiency	All	Enema	77.59 ± 13.77	75.26 ± 14.84	80.05 ± 12.27	0.130
		Nelaton catheterization	81.11 ± 12.94	79.62 ± 12.24	82.68 ± 13.63	0.306
		Nasogastric feeding	81.97 ± 14.45	82.08 ± 13.28	81.86 ± 15.78	0.949
	Core	Enema	32.67 ± 5.53	32.00 ± 5.95	33.38 ± 5.04	0.281
		Nelaton catheterization	48.79 ± 7.86	47.59 ± 7.56	50.05 ± 8.09	0.174
		Nasogastric feeding	41.01 ± 7.45	40.89 ± 6.84	41.14 ± 8.15	0.889
Learning Satisfaction			4.27 ± 0.57	4.18 ± 0.60	4.36 ± 0.53	0.167
Task Engagement		Self-regulation	5.38 ± 0.97	5.23 ± 0.94	5.54 ± 0.98	0.171
		Experiencing positive emotions	5.52 ± 0.96	5.36 ± 1.00	5.70 ± 0.90	0.127
		Attention on task	5.52 ± 0.91	5.23 ± 0.91	5.82 ± 0.83	0.004 *
		Overall	5.46 ± 0.84	5.27 ± 0.81	5.66 ± 0.83	0.039 *

* *p* < 0.05.

**Table 3 healthcare-12-01109-t003:** Differences between groups in post-test scores after covariation of pre-test scores.

Variables			Group	Pre-Test	Post-Test	95% CI	T (*p*1)	F	Sig (*p*2)
				M ± SD	M ± SD	(Lower–Upper)			
Nursing Performance Confidence			Cont. (n = 37)	14.18 ± 4.10	14.79 ± 3.82	0.62(−1.03, 2.26)	0.453	3.878	0.053
			Exp. (n = 39)	17.11 ± 3.81	17.94 ± 4.79	1.06(−0.42, 2.53)	0.155		
Proficiency	All	Enema	Cont.	75.26 ± 14.84	78.13 ± 15.74	2.87(−1.58, 7.33)	0.200	0.110	0.741
			Exp.	80.05 ± 12.27	86.05 ± 14.81	6.00(1.27, 10.73)	0.014		
		Nelaton catheterization	Cont.	79.62 ± 12.24	74.69 ± 16.54	−4.92(−9.58, −0.26)	0.039	2.680	0.106
			Exp.	82.68 ± 13.63	86.08 ± 13.46	3.41(−1.83, 8.65)	0.196		
		Nasogastric feeding	Cont.	82.08 ± 13.28	76.05 ± 15.45	−5.82(−9.63, −2.00)	0.004	6.581	0.012 *
			Exp.	81.86 ± 15.78	84.73 ± 13.82	2.72(−3.16, 8.61)	0.354		
	Core	Enema	Cont.	32.00 ± 5.95	31.95 ± 6.65	−0.05(−2.10, 2.00)	0.960	0.073	0.787
			Exp.	33.38 ± 5.04	34.65 ± 5.96	1.27(−0.73, 3.27)	0.205		
		Nelaton catheterization	Cont.	47.59 ± 7.56	44.56 ± 10.21	−3.03(−5.84, −0.21)	0.036	4.046	0.048 *
			Exp.	50.05 ± 8.09	51.54 ± 7.98	1.49(−1.77, 4.74)	0.361		
		Nasogastric feeding	Cont.	40.89 ± 6.84	38.03 ± 7.73	−2.76(−4.86, −0.67)	0.011	5.389	0.023 *
			Exp.	41.14 ± 8.15	42.51 ± 7.03	1.31(−1.83, 4.44)	0.404		
Learning Satisfaction			Cont.	4.18 ± 0.60	4.21 ± 0.65	0.03(−0.19, 0.26)	0.759	0.668	0.416
			Exp.	4.36 ± 0.53	4.50 ± 0.56	0.14(−0.03, 0.30)	0.110		
Task Engagement	Self−regulation		Cont.	5.23 ± 0.94	5.45 ± 1.16	0.22(−0.20, 0.63)	0.294	0.003	0.954
			Exp.	5.54 ± 0.98	6.22 ± 0.73	0.68(0.37, 0.99)	<0.001		
	Experiencing positive emotions		Cont.	5.36 ± 1.00	5.20 ± 0.91	−0.16(−0.59, 0.26)	0.444	2.187	0.144
			Exp.	5.70 ± 0.90	5.97 ± 0.89	0.27(−0.06, 0.60)	0.109		
	Attention on task		Cont.	5.23 ± 0.91	5.37 ± 1.18	0.14(−0.27, 0.55)	0.487	0.309	0.580
			Exp.	5.82 ± 0.83	6.29 ± 0.78	0.47(0.22, 0.72)	0.001		
	Overall		Cont.	5.27 ± 0.81	5.36 ± 1.03	0.09(−0.27, 0.44)	0.622	0.164	0.687
			Exp.	5.66 ± 0.83	6.17 ± 0.75	0.50(0.25, 0.76)	<0.001		

Note: *p*1: *p*-value for the comparison of pre–post differences (by paired *t*-test); *p*2: comparison result for the difference (*p*-value of interaction term using ANCOVA); * *p* < 0.05. Note: Cont. = control group; Exp. = experimental group.

## Data Availability

The data presented in this study are available on request from the corresponding author. The data are not publicly available due to domestic law.

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
