# Peer review of "Virtual Reality Simulation-Based Clinical Procedure Skills Training for Nursing College Students: A Quasi-Experimental Study"

_healthcare, 2024, doi:10.3390/healthcare12111109_

Round 1

Reviewer 1 Report

Comments and Suggestions for Authors

Dear Authors

It has been a pleasure to have the opportunity to review your manuscript “Virtual reality simulation-based clinical procedure skills training for nursing college students: A pre-post experimental design”.

Here are some recommendations that may help to improve the manuscript:

LINE 40-42: “Core nursing skills are essential as they are frequently used in the daily responsibilities of nurses and constitute a fundamental component of nursing education and training  [5,6]”. Core nursing skills are important, but the student must achieve competencies to develop these basic skills. The acquisition of knowledge by the students allows them to develop the most demanding basic techniques or procedures, in addition to reinforcing skills and attitudes through testing. All the above allows the future professional to provide excellence in care. It is recommended to expand the statement (40-42) to include all aspects that influence the education of nurses.

LINE 84-86: “We aimed to evaluate the effectiveness of 3D VR education by implementing three IVR programs covering enema, nasogastric feeding, and nelaton catheterization.” AND LINE 90-93: “This study employed a non-equivalent control group pre-post experimental design to  assess the impact of an IVR program on nursing students’ performance confidence, proficiency, task engagement, and learning satisfaction while focusing on core nursing skills  such as enema, nasogastric feeding, and nelaton catheterization.” Please review the objective and type of study, perhaps the objective needs to be adjusted to the type of study, there are more variables to evaluate than satisfaction (confidence, competence, commitment, and satisfaction with learning).

LINE 183-184: “The proficiency assessment instrument comprised 10 questions, including each core performance item specified for core fundamental nursing skills”. Regarding the 10 questions of the questionnaire, does the document you refer to also have 10 items? If the reference document has more than 10 questions, why were 10 selected and what was the selection strategy?

LINE 206: The data were analyzed using SPSS PC+ 22.0 for Windows. Add (Armonk, NY, USA).

LINE 336-338: “The study results demonstrate a significant increase in core nursing skills among students who underwent this VR training.”  I think that in this statement they should be clear, specifying in which variables there has been an increase (confidence, competence, commitment, and satisfaction with learning) and that only in one of them (competence) a significant relationship was found.

I recommend making minor revisions.

Wish you the best of luck with your manuscript.

Greetings

Reviewer 2 Report

Comments and Suggestions for Authors

Dear Respectable Authors

Thank you for considering a great area of research related to nursing education and nursing students. You conducted a virtual reality simulation-based clinical procedure skills training for nursing college students using a pre-post experimental design. Your results are of interest but your manuscript needs some revisions as follows;

- Abstract, please add a clear aim of the study. 

- Abstract, please add study design, sampling methods, and how to allocate the participants to the groups.

- Abstract, VR stands for what?

- Line 21, R stands for what?

- Abstract, please add some demographic information including mean age and gender percentages overall or based on different groups.

- Please add statistical results (level of significance) to the abstract including P=value or/and CI.

- Line 24, C stands for what?

- Please add the reasons for the exclusion of the participants in each group.

- Your discussion section needs promotion. You need to discuss all your results with the results of the other researchers. As the number of your results and domain are frequent you need to discuss more. The number of citations used for comparison is not enough. Your results are interesting and it is a pity that it is not compared with other similar studies. These comparisons can give researchers new perspectives. In addition, based on your results, provide some practical suggestions for future research. These suggestions should cover the limitations of your study so that we can achieve more solid results.

Cheers
